# Application Layer Packet Processing Using PISA Switches

**DOI:** 10.3390/s21238010

**Published:** 2021-11-30

**Authors:** Ismail Butun, Yusuf Kursat Tuncel, Kasim Oztoprak

**Affiliations:** 1Department of Computer Engineering, KTH Royal University of Technology, SE-114 28 Stockholm, Sweden; 2Department of Computer Engineering, Konya Food and Agriculture University, Konya 42080, Turkey; yusuf.tuncel@gidatarim.edu.tr (Y.K.T.); kasim.oztoprak@gidatarim.edu.tr (K.O.)

**Keywords:** software-defined networks, protocol independent switch architecture, programmable switches, P4, virtualization, stream processor, deep packet inspection

## Abstract

This paper investigates and proposes a solution for Protocol Independent Switch Architecture (PISA) to process application layer data, enabling the inspection of application content. PISA is a novel approach in networking where the switch does not run any embedded binary code but rather an interpreted code written in a domain-specific language. The main motivation behind this approach is that telecommunication operators do not want to be locked in by a vendor for any type of networking equipment, develop their own networking code in a hardware environment that is not governed by a single equipment manufacturer. This approach also eases the modeling of equipment in a simulation environment as all of the components of a hardware switch run the same compatible code in a software modeled switch. The novel techniques in this paper exploit the main functions of a programmable switch and combine the streaming data processor to create the desired effect from a telecommunication operator perspective to lower the costs and govern the network in a comprehensive manner. The results indicate that the proposed solution using PISA switches enables application visibility in an outstanding performance. This ability helps the operators to remove a fundamental gap between flexibility and scalability by making the best use of limited compute resources in application identification and the response to them. The experimental study indicates that, without any optimization, the proposed solution increases the performance of application identification systems 5.5 to 47.0 times. This study promises that DPI, NGFW (Next-Generation Firewall), and such application layer systems which have quite high costs per unit traffic volume and could not scale to a Tbps level, can be combined with PISA to overcome the cost and scalability issues.

## 1. Introduction

The telecommunication world is undergoing a great transformation. The most important aspect of this transformation is to switch from old hardware-dependent, vertical architectures to software-defined architecture. Although the use of NFV was a key improvement in a data plane with improved flexibility, Protocol Independent Switch Architecture (PISA) is one of the key elements with the accelerated performance and intelligent processing ability in the data plane during this change. The change in the architecture affects all stakeholders in a telecommunication operator infrastructure including applications. Legacy Applications written for legacy hardware are transformed into software-defined architecture.

Independent from the Software Defined Architectures, application identification became critical in the last decade. It positioned itself in the center of cyber-security, accounting, quality of service management, and similar services. One of the most important problems incurred by application identification is resource hungry behavior in itself. Next-Generation Firewalls (NGFW), and Deep Packet Inspection (DPI) systems are two of the most popular usage areas of application identification. DPI, as the name implies, inspects every packet that is running through the network deeply, and tries to classify it under a human-readable name. It not only relies on packets metadata and headers but also packet payload, hence the name “Deep”.

While L4 (OSI Layer-4) provides valuable information about a packet, it cannot give us any clue about the payload. In order to do that, packets must be inspected by maintaining the stateful information, and the payload must be constructed accordingly, so that it can be classified correctly [1]. With the help of L4 information, network-side security, such as stateful firewalls, can be built. Similar to NGFW processing packets in L7, DPI still needs to be inspected at L7. With the emergence of SDN architecture, DPI vendors switched from hardware to software based L7 DPIs. As they switch from hardware-dependent architecture to SDN based architecture, they lack the proper scalability to match the actual line speed of the switches. While the capacities of the data backbone increase, the systems depending on application identification became the bottleneck of the infrastructure.

In this transformation, the network applications become a Virtual Network Functions (VNF). Current state-of-the-art high performance software-based DPI systems (DPI VNFs) can scale up to 160 Gbps in a Virtual Machine running on top of powerful hardware [2]. As the demand increases, the telecom operators will need application identification systems like NGFWs and DPI systems running with the speeds in the order of Tbps of traffic classification in real time and such as in a single instance of DPI. The performance gain arises from the fact that the classification operation starts at the switch-level code data plane and continues in the user-plane.

In the meantime, several parallel works concentrated on various aspects (machine and deep learning methods, cybersecurity, etc.) of SDN. For instance, Ref. [3] argued about the necessity of providing quality of service (QoS) for each application on the network by the network operators, which can be accomplished by classifying network traffic associated with the applications. Authors have shown that deep learning models can be employed for classifying the network traffic, and residual network (ResNet) model outperforms the CNN convolutional neural network (CNN) model. In another work, cybersecurity related issues on the network layer are investigated [4]—for instance, detecting application-layer DoS attacks that utilize encrypted protocols by applying an anomaly-detection-based approach to statistics extracted from network packets.

In this paper, we aimed to introduce application layer processing capabilities of P4-based programmable switches and their usage in application layer processing. We investigated and proposed a solution for Protocol Independent Switch Architecture to process application layer data, enabling the inspection of an application content and triggering appropriate response. Protocol Independent Switch Architecture is a novel approach in networking where the switch does not run any embedded binary code but rather an interpreted code written in a purpose-specific language. The main motivation behind this approach is that telecommunication operators do not want to be locked in by a vendor for any type of networking equipment, developing their own networking code in a hardware environment that is not governed by a single equipment manufacturer. This approach also eases the modeling of equipment in a simulation environment as all of the components of a hardware switch run the same compatible code in a software model. The novel techniques in this paper exploit the main functions of a programmable switch to create the desired effect from a telecommunication operator perspective to lower the costs and govern the network in a comprehensive manner.

As stated before, the current demand in traffic growth puts a burden on the applications running in the application layer in the telecommunication world, although the performance and capacities of DPI systems and Next-Generation Firewalls do not grow with the demand of traffic growth; in addition, they cannot adapt themselves to the current revolution which migrates the networks into SDN-based systems. This paper proposes a solution using PISA switches with a DPI enabling application visibility (type identification) in an outstanding performance. The proposed architecture processes the packets in a network switch while selecting only necessary ones to the L7 based systems such as DPI and NGFW. The proposed solution distributes a load of DPI/NGFW systems into PISA switches and DPI/NGFW systems. Using the proposed solution in a network allows the users to grow in the Tbps scale as well as benefit from Network/Service Function chaining, which will also remove the overhead of passing through all inspection systems for unnecessary traffic. The simulation studies demonstrate that this approach increments the performance of NGFW and DPI systems in the order of 40 times. Building such a flexible and scalable application visibility system is challenging. This study also tries to give an answer to how network operators should design their networks in order to benefit from such solution processing packets in L7 knowledge with the performance of L4; in other words, they should figure out how to scale out such system for a high volume of data in real time.

## 2. Background

### 2.1. Protocol Independent Switch Architecture (PISA)

The research on programmable switches led to the definition of a re-configurable match-action table (RMT) [5] based hardware that can be programmed with a domain-specific language. Protocol Independent Switch Architecture (PISA) is a special case of RMT that supports the P4 language as the default domain-specific language [6].

A typical PISA switch consists of a programmable parser, ingres match-action table, a queue, a set of registers to keep the state of variable, egres match-action table, and a programmable deparse as shown in Figure 1.

The parser and deparser are programmed for processing user-defined packet header formats. The ingress and egress pipelines are the actual packet processing units that go through match-action tables in stages. Match-action tables match the header based on a set of rules that is controlled by control plane and performs the corresponding action on the packet. Actions use primitives to modify the non-persistent resources (headers or metadata) of each packet.

### 2.2. P4 Language

Although there are several studies developing and using programmable hardware [8,9,10,11], the early use of programmable hardware is to make telemetry data easy to use. Telemetry data are crucial for an automated future but generating telemetry data is not a trivial task. Adding more hardware and software to the routing and switching systems makes the current architecture more complex than ever. Since the telemetry data are generated at the packet level, the most logical way of doing this seems to be arising from the packet generating software at the hardware level, which leads us to P4, Programming protocol-independent Packet Processors, as it is referred to in the original paper defining it [12]. P4 is a domain-specific programming language for packet-processing hardware such as a router, switch, network interface cards, and network function related appliances that work and data plane based on the decisions from the control plane as in Figure 2.

In a typical PISA switch, execution of a P4 program is explained in Figure 3.

The user develops a P4 program, which can be any type of network function, such as router, firewall, load balancer, or packet inspection switch.P4 compiler compiles the program as a JSON file and sends it to the switch, which can be a physical switch or a software model of it.The states of parser, match-parser, match-action tables, ingress, egress queue, and deparser is controlled by P4 execution.The states of match-action tables are additionally controlled by control-plane which can change the behavior of the P4 code at run-time.

When the P4 compiler is placed between the program and the API, the P4 compiler translates the domain-specific language P4 code into a JSON file, which acts as an executable file for the PISA switch. The required CLI commands to configure to be switched are also sent with this JSON file, which typically contains the newly added match-action table names, ingress, and egress queue names to be created on the PISA switch. This JSON file is actually a series of match-action table entries that acts as an executable for the switch to change the state of its tables based on the incoming packet.

The control plane commands contains the necessary table initialization based on the packet processing actions. The implementation of P4 control plane commands may differ from each other depending on the switch type (physical or virtual), vendor and the version of P4 (P4-14, P4-16). The following commands are valid for Simple Switch Behavioral Model V2, P4-16: [14] table_set_default <table_name> <action_name> <action_parameters>is used to set the default action (i.e., the action executed when no match is found) of a table.table_add <table_name> <action_name> <match_fields> => <action_parameters>is used to set the action related to a specific match in a table.mirror_add <source> <destination>is used to mirror a specific port internally.

P4 programs ease the development of a network equipment code to a level for which only 128 lines are enough to build a simple IP switch with header validation [15]. Although the language itself is simple, there are other tools that emit P4 language code from another high-level language, such as the work done by the authors [13], P4HDL, which generates P4 code from a pseudo-code.

### 2.3. In-Band Telemetry with Programmable Switches

The above three requirements to develop a programmable hardware are not the only features addressed by P4. One of the most promising features of P4 arises in the telemetry. In-band Network Telemetry (INT) is defined in P4 language as one of the main applications [14]. Since P4 executes at the packet-processing level, it can rewrite every segment of the packet header, including the custom headers. This type of modification cannot be done in traditional statically programmed hardware-based network equipment. P4 helps set up a data plane by using the packet headers appropriately to collect even more information on the network’s status than what we can determine using conventional methods [16]. The idea behind INT is to collect telemetry metadata for each packet, including routing paths for the packet, entry and exit timestamps, the packets’ latency, queue occupancy in a given node, use of egress port connections, and the like. These measurements can be produced by each network node and sent in the form of a report to the monitoring system. Another way to embed them in packets is to delete them into allocated nodes at any node on the packet visits and connect them to the monitoring system. In a recent study, researchers used P4 INT experimental validation for telemetry based monitoring applications on the multi-layer optical network switches [17]. Using the telemetry data and the integrated software around it, semi-automatic congestion control over optical network switches can be achieved with the currently available SDN/NFV systems.

Although telemetry data can be collected in any way that is defined by P4 code, there are two types of telemetry that are defined in a standard P4 implementation [6]. As shown in Figure 4, telemetry data can be either embedded within a packet, which is called INT-MD, or extracted as a separate packet, called INT-XD. INT-MD is usually used by intermediate routers (switches to identify any type of problem that might occur along the path, which INT-XD is useful for external applications that do not need the payload of the original packet. In this experiment, INT metadata are used to help to the measurement of a switch internal state such as ingress/egress port ID, switch ID, queue occupancy, processing time, etc. These metrics are application agnostic and help in application-layer processing.

### 2.4. Real-Time Data Streaming

Real-time data streaming is shown to be beneficial for safety critical networks by removing possible bottleneck situations at the data accumulation joints, such as the data aggregator switches at the industrial networks [18]. In these networks, a possible delay in data would cause disastrous events, and data-streaming is a very good candidate solution as a remedy to this problem [19]. In the context of programmable switches, real-time data streaming is combined with telemetry to add application analytics, visibility, and troubleshooting features to a network stream. Apache Spark [20] and Apache Flink [21] are two of the most prominent pieces of software that is being used in streaming network telemetry data.

### 2.5. Deep Packet Inspection (DPI) and Application Layer Visibility

Deep Packet Inspection is important for telecommunication operators to gain more insight about the network and subscribers for revenue generation as well as cyber-security. A series of research [22,23] made in this area by the same author showed that subscriber profiling based on application level classification is critical for operators to increase the revenue and generate insight about the network. As the name implies, DPI inspects every packet with respect to the source, destination, header information, payload, and any other layer that is wrapped into it. Application layer visibility enable operators to distinguish between their subscribers and offer them new subscription services accordingly. As the video content is on the rise, operators can offer subscribers based on their use of online video services, such as Netflix, Amazon Prime, or Hulu. In addition, DPI is a supportive tool in employing Lawful Intercept or applying some appropriate filters to the Internet access of children.

## 3. Application Layer Processing with P4 Switches

The transformation from legacy systems into software defined architectures triggered the change in the hardware architectures. The demand for the change resulted in the development of PISA switches. The current state of the art in a PISA switch can scale up to 12.8 Tbps with a single ASIC/FPGA interface running with the speed of 400 Gbps. After the introduction of PISA switches into the production environment, the applications running in L4 such as Load Balancers, Volumetric DDoS attack detection, and prevention systems, port-based DNS applications are being ported into PISA switches. In this study, we aim to extend the use of PISA switches into L7 applications by designing a proper architecture. In the proposed architecture, by using PISA switches and its primary programming language P4, an application-level traffic analyzing system is proposed in a software-based emulation environment. It is basically combining L4 analytics of P4 architecture and L7 properties of the current state of the art in DPI or similar application layer inspection systems. The proposed architecture can be used to build a brand-new NGFW or DPI, by eliminating the complexities arising from switch dependent code.

### 3.1. Proposed System Architecture

The proposed system architecture in Figure 5 consists of five main components: PISA, Stream Processor, Control Plane, and Data Plane Configuration.

PISA: Programmable Switch that can run multiple instances of different P4 codes.

Data Plane: The generated P4 code for specific monitoring/telemetry/DPI/NGFW tasks. These P4 programs can be deployed according to specific task needs.

Control Plane: Programmable Switch related control plane engine to be placed. The control plane is aware of Data plane drivers and can communicate with the underlying switch according to the specific tasks. Although the proposed architecture supports any application specific task, from now on, the architecture will be coupled with DPI use case to make it easier to understand. This module is DPI-aware, which is fed from the specific packet stream, so that any decision to be made on the switch can be controlled by examining the specific packets.

Stream Processor: The stream processor to operate on the matching stream patterns based on the decisions taken from data plane configuration. Specific telemetry tasks can be offloaded to the stream processor to decrease the workload over the switch or vice versa. Workload trade-off between the stream processor and the switch is based on the number of streams that matches a specific monitoring task.

Application-Level Visibility: Application-level visibility is the component that actually identifies the types of application based on their L4 to L7 properties, which is also called DPI. In a typical DPI system, a server with network interfaces is running the DPI application. There are two usage modes of DPI systems which are active and passive DPI systems. In the passive mode, they are fed by a mirror of the traffic and processes it offline. On the other hand, active DPI systems fall within the whole traffic and are supposed to process all the traffic piece by piece in real time.

In the proposed architecture, the PISA switch processes the packets in the network layer and can even process the flows in the transport layer and co-operates with the stream processor to identify the applications. This is the point where the aggregation–disaggregation of high performing PISA switch and application identification engines.

The PISA switch selects the minimal packets from the flows and forwards them to the stream processor/DPI engine to identify the applications and generate the actions among the predefined policies. The proposed architecture combines the power of PISA and L7 application inspection/classification/processing features by designing them together. The simulation results indicate that in the near future most of the systems using application awareness will re-design their systems running on top of PISA switches together with their redesigned applications as a stream processor. The following algorithm shown in Box 1 explains our approach:

Listing 1Pseudo Code proposed for the P4 switches.
While packet -> in ingres buffer
	Extract telemetry headers
	Put in Flow-Keys Telemetry Headers
	If Flow Not in Flow-Table
Create flow in Flow-Table
	Else IF Flow-Packet-Count.< 2
		Put Payload in Flow-Packets ...
		with Flow-Keys in Flow-Table
Continue
	Else
		Create telemetry header with ...
		    INT-XD options
		Send Flow-Table in Flow-Keys ...
		    to External Telemetry	   


The accurate accounting of the flows can also be done with P4 language. The accounting of a flow should include the following information: Considering the definition of the flow, for every flow, count number of packets, number of bytes, flow start time, flow end time, in addition to that, for TCP flows, TCP flags.

The P4 code on a switch would combine the accounting information and send the rest to the aggregator with a pseudo-code. Please see the Appendix A for the details of the mentioned pseudo-code.This pseudo-code works as the preprocessor of the flow, extracts the required fields and sends it to the stream processor for further processing.

Lastly, the traditional DPI systems has two operating modes:InlineOut-of-Band

In the inline mode, DPI systems are placed between the edge and core network, so that the traffic is processed as the flow continues. This operating mode enables DPI to apply policies directly on the flow, without requiring any other hardware. The biggest disadvantage of this approach is that the DPI becomes the weakest link of the network, it should be scaled at least as much as the aggregated sum of the traffic received from the edges.

In out-of-band mode, DPI acts like a simple traffic analyzing tool; it received the traffic passively from a mirror port of a network aggregation device, collecting all the traffic information and applying policies accordingly. In this mode, the biggest challenge is policy application, as the traffic is not directly passing through the DPI; it can only act on TCP traffic by sending TCP-resets to the source addresses, for example, to apply a restricted access policy to a particular destination address within the scope of the network. Other types of policy applications, such as bandwidth restriction, quality-of-service changes, etc., require control plane integration with the underlying network device.

Our architecture also combines the benefits of inline DPI devices with the out-of-band ones where the traffic is actively received on the switch, counted and reported on the aggregated external devices and the policies are actively applied as the event triggers occur.

### 3.2. Simulation Environment

To simulate the proposed architecture, the following components are built as a development and simulation environment:

P4 Simulation Environment: This is the default simulation target for BMV2 PISA switches, as shown in Figure 6, which includes Mininet by default and handles virtual NIC creating, switch port allocation, connecting the switch port to the host process, and running the rest of the packet flow.

Virtual Machine: This is the default virtual machine, built in a programmatic way with Vagrant, a developer friendly VM running environment, based on Ubuntu 14.04 (ubuntu/trusty64) and several other necessary components, as shown in Box 2:

Listing 2Virtual machine set-up.
Simple_switch_bmv2: 
BMV2 software switch, based on Python2.7
	m-veth-1: Ingres mininet Switch Port
	m-veth-2: Egres mininet Switch Port
	out-veth-1: Ingrest Server Host Port
	out-veth-2: Egres Server Host Port		


## 4. Experimental Study

In this experimental study, we have used a P4 simulation environment which was discussed above and presented in Figure 6. The following items describe each component of our experimental simulation environment in detail:

Flow Generation: This is the controlled flow generation tool, written in Go. Synthetic flows are created with Python, while real-flow is taken from the Canadian Institute for Cyber-Security [24].

DPI: Deep Packet Inspection module written in Go, based on nDPI [25].

Emitter: Flow emitter that reads from the mirroring port, extracts metadata header information written by Data-Plane and sends the rest of the packet for stream processor. This module is also Apache-Spark aware; the final result of the telemetry query is calculated by the Emitter module.

Stream Processor: The streaming processor for the rest of the flows that match the final criteria for the expected output. In this simulation, we used Apache-Spark as a stream processor. The stream processor will be upgraded to Apache-Flink for better scalability.

Switch Script Control: This script controls the switch tables to update the relevant switch tables under control.

The parameters for running the simulation are adjusted according to the following criteria:Session is TCP (Session has 3-way handshake);Session is UDP (Session has no 3-way handshake);Session is detected by nDPI;Session is not detected by nDPI.

### 4.1. Experiment-1: Application Identification Performance Improvement DPI Application Classification on Mixed Flow Captures

Our hypothesis is that, to identify an application in a packet, a few bytes in a flow (one or two packets depending on the application) should be enough to determine the type of application correctly. Keeping this in mind, we must first identify the session in a packet. This use case demonstrates the performance improvement in DPI systems by eliminating the number of packets by some factor.

In order to adjust the parameters of this identification, we first analyzed the packet stream with nDPI, counting the number of identified protocols and the number of packets that are included in each stream. We then start reducing the number of packets in each stream and run the protocol identification with nDPI once again, comparing the results of identification with the previous run. By reducing the number of packets each time, we calculated the number of identified protocols in each reduced packet stream.

Session Identification in an IP flow is based on two different IP sessions:

#### 4.1.1. TCP Session


*SrcIP, DstI, SrcPort, DstPort, TCPSeqNum*


TCP Session Identification is based Source IP, Destination IP, Source Port Destination Port and the TCP Sequence Number. The TCP session is established after the 3-way handshake as shown in Box 3:

Listing 3Pseudo Code proposed for the 3-way handshake.
---
Source -> Destination (SYN+Seq #)
Destination -> Source (SYN ACK+Seq #)
Source -> Destination (ACK+Seq #)
---


After the last ACK of the source, Sequence Number is incremented for a flow in the TCP session. Actually, it comes from the nature of TCP. It starts randomly and increments by the amount of the data transferred in each packet. The same is valid for ACK number.

The packets that will be reduced should be the packets after this 3-way handshake packet. To identify the flows, we will use the packet SYN ACK, and the response to the third packet—in other words, the first two packets of the server (or destination to source).

#### 4.1.2. UDP Session


*SrcIP, DstI, SrcPort, DstPort*


UDP is a connectionless protocol; there is no clear definition of a UDP session. Every packet may create a flow independently. Basic identification for UDP flow consists of Source IP, Destination IP, Source Port, and Destination Port. Since Source Port is randomly allocated depending on the OS (which is called ephemeral ports), any flow that is using the same source port is considered as the same UDP session.

### 4.2. Sample Packet Captures

To study the flow reduction, we used the sample captures from nDPI that is used for verification of protocol identification. The capture files consist of 183 files, containing more than one protocol in one capture file. Twenty-two files that are too small for reduction (having packets less than 2) are excluded from study. One packet especially crafted for testing invalid packet type is also excluded since we are interested in valid packets, leaving us 160 packet captures.

To reduce the flow, the following pseudo-code is used as shown in Box 4:

Listing 4Pseudo Code proposed for the reduced algorithm.
---
network_packets = rdpcap(infile) 
sessions = network_packets.sessions()
for key in sessions:
        pktCount=0
        for pkt in sessions[key]:
                if (pktCount < 2):
                        write(pkt, outfile)
                        pktCount = pktCount + 1

---


In this code, sessions are extracted by the criteria of whether they are TCP or UDP sessions. As mentioned earlier, for the TCP session, 3-way handshake packets are excluded from the session, whereas, for a UDP session, there is no precondition to exclude the packets. We use the second packets of the 3-way handshake as the first packet of the flow.

After the extraction of sessions, an nDPI sample classifier is used to classify the application in each reduced capture by replaying the capture file on the switch.

The following Table 1 summarizes the results of the experiment:

The full data are available in Table A1.

### 4.3. Experiment-2: TCP-Based Application Identification Using Real-Life Data

In the second experiment, we used the real captures from [24], namely the files in the dataset named PCAP-01-12_0750-0818.

There are 69 files located in this dataset, each containing a real world data capture that contains data from a real DDoS attack along with different types of traffic.

To see the effect of TCP, we extracted the TCP streams and used the extracted streams to send to the simulation.

For the sake of convenience to the readers, the results in Table A3 are summarized in the following Table 2:

### 4.4. Experiment-3: Application Identification in Full Stream Using Real-Life Data

In the final experiment, using the same capture files in Experiment-2, we treated the streams as is, sending them directly to the switch. The following results in Table 3 are achieved.

Full results are given in Table A2.

### 4.5. Results and Discussion of the Experiments

The experimental study on the packet captures showed us that 2-packet reduction of a flow is accurate enough to identify a flow.

In Experiment 1, the decrease in detection rate is mostly caused by TLS encryption, which shows us that further study is needed to identify encrypted flow as shown in Table 1. An ML based approach would be implemented to success in application identification of all flows. Based on the results from Table A1, 125 out of 160 packet captures are correctly identified. Sixteen out of 160 packet captures could not be identified. Normally, 160 out of 160 packets would be identified correctly. In addition, 125 files were identified correctly; 16 not identified at all (0 identification); 19 partially identified; 16 non-identified protocols are completely encrypted protocols; 125 identified protocols, mixed partially TLS and plain protocols; and 19 partially identified protocols are mixed partially TLS and plain protocols. Detection Rate drops with the reduced flow. (i.e., as we reduce the flow, we also lose important flow information that is needed for packet identification, short flows) The reason for not identifying these packet captures are that they are mostly encrypted protocols, which require more than two packets to identify. We will expand the experiments according to this. Since the flows used in Experiment 1 are taken from nDPI’s test captures, they consist of an artificially selected short flow containing all of the applications that nDPI can identify.

In Experiment 2, the results in Table 2 showed that it is possible to increase the detection rate while the reduction rate is also increased. This is due to the fact that there are only 17 protocols detected in TCP streams as indicated in Table A3, and most of them are not TLS-based protocols, or can be identified without deep inspection of the remaining payload.

In Experiment 3, the results in Table 3 indicated that, if we include UDP streams, the accuracy goes even higher, but the reduction rate decreases. This behavior is expected since the number of detected applications in Table A2 is 84, more than the number applications detected in TCP streams, but the number of packets in UDP streams is lower than the number of packets in TCP streams. This result is in line with results obtained from the study in [26].

## 5. Conclusions

The results of this study indicate that the application layer data processing can be done with PISA switches. We do not always need complex techniques to inspect the packets in L7, and a simple flow-based packet reduction can achieve significant accuracy to identify the flows and add application-level visibility over the network. Streaming processing combined with switch-level applications helps us build strong networking applications. In-band Network Telemetry is in the central position of a programmable switch that distinguishes and separates them from the traditional switches. The proposed method constructs a Network Processor with a specific task from each PISA-stream processor pair. The simulation results indicate that using such PISA switches in the center of all network traffic will increase the performance of such systems on the order of tens of folds. The use of such (proposed) systems will solve the capacity problems experienced with applying full network service chaining. In other words, by using a single PISA switch and tens of stream processors with different features (DPI, NGFW, etc.) on different ports, it constructs a big traffic exchange fabric with dynamically attached Network Processors of different types with very low costs.

The results of this study demonstrate that the proposed system reduces the traffic load of such systems by a factor of 5.5 to 47.0 times with acceptable application identification. Applying some ML based approaches would increase the success rate as if all traffic is going through legacy systems with the higher power of proposed systems. In addition, real traffic scenarios indicate that the performance gain would reach up to a factor of 40 on average by using the statistics in this study [26].

The studies in the literature and our experimental studies demonstrated that PISA switches are the glue for the SDN-NFV couple increasing the performance of such systems. One of the major problems of the NFV based application layer processing systems were the network packet processing performance bottleneck; however, the proposed solution offering an architecture avoids the performance bottleneck of both PNF (Pyhsical Network Function) and VNF (Virtual Network Function) systems by decreasing the network packet load.

## 6. Future Study

DPI, NGFW (Next-Generation Firewall), and such application layer systems that have quite a high cost per unit traffic volume and could not scale to a Tbps level can be combined with PISA to overcome the cost and scalability issues. Practical applications are expected to be available in the upcoming years, maybe even months.

Encrypted network traffic identification with P4 language is one of the main future areas of study for this thesis. In-band Telemetry seems to be a good place to start this study, as it tells us about the characteristics of a flow on a packet level. In this kind of an analysis, AI/ML methods can provide a lot of help in defining the features of traffic. As stated above, the use of PISA switches will allow the operators to collect in-band telemetry information, which will also create the building ground for Zero Touch Networking (ZSM) once the networks are utilized with the use of proposed systems. Once ZSM features are injected into the infrastructures, operational costs and outage times will decrease dramatically.

Another area of interest based on this study could be Digital Twins in Telecommunication Networks. As PISA switches allow you to model the hardware in a software environment, it would be straightforward to build a DT (Digital Twin) of a telecom network and feed forward the actual data and commands towards the active network. Particularly, the data center network can be modeled completely using the DTs of core and edge network devices. Telcos can gain an advantage from this by running different scenarios on their DT based on different types of network flows. These network flows can be adjusted to plan the data center network topology according to the SLA of the customers.

## Figures and Tables

**Figure 1 sensors-21-08010-f001:**
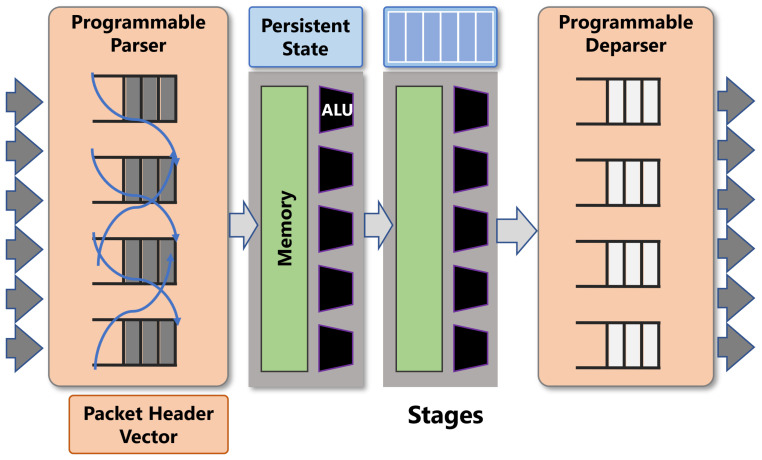
PISA match-action table processing pipeline (Source: Adapted from [7]).

**Figure 2 sensors-21-08010-f002:**
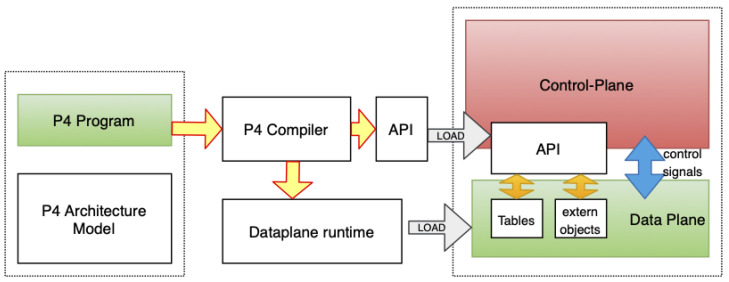
P4 Architecture (Source: Adapted from [9]).

**Figure 3 sensors-21-08010-f003:**
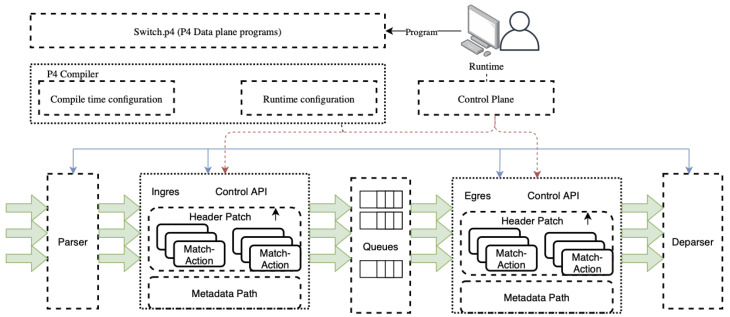
Pipeline execution in a P4-enabled switch (Source: Adapted from [13]).

**Figure 4 sensors-21-08010-f004:**
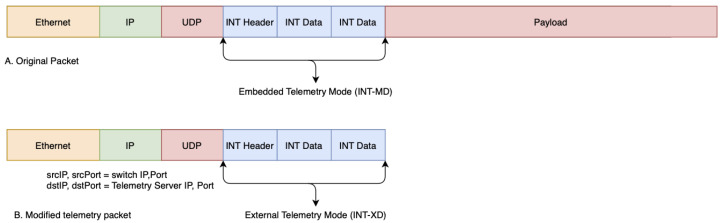
In-band network telemetry.

**Figure 5 sensors-21-08010-f005:**
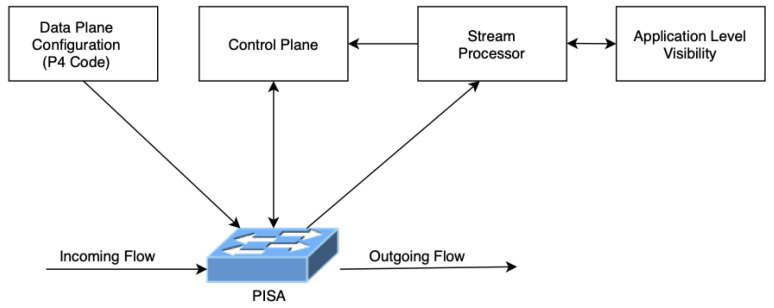
Proposed system architecture.

**Figure 6 sensors-21-08010-f006:**
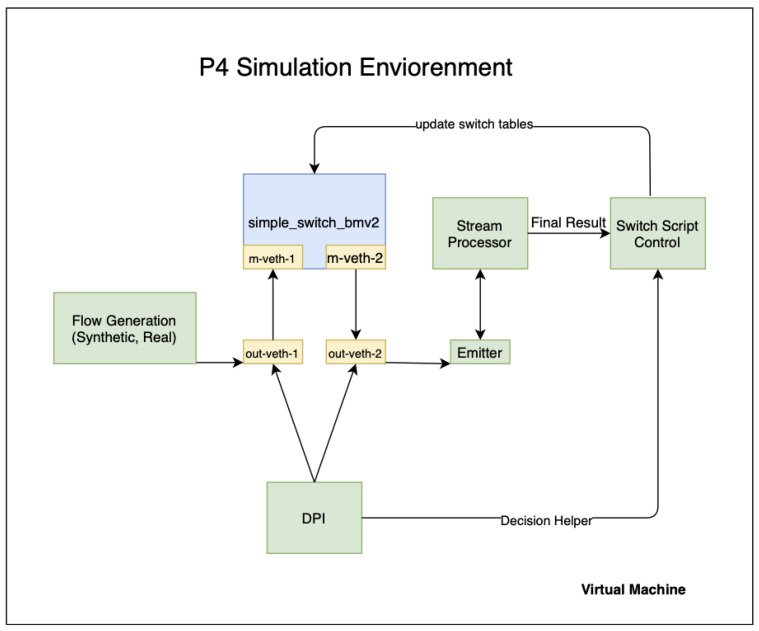
Simulation environment.

**Table 1 sensors-21-08010-t001:** Rates for Test Captures.

μ REDUCTION RATIO	82%
μ REDUCTION FACTOR	5.5
μ DETECTION RATE	84%

**Table 2 sensors-21-08010-t002:** Rates for real-life captures using only TCP streams.

μ REDUCTION RATIO	97.88%
μ REDUCTION FACTOR	47.16
μ DETECTION RATE	95%

**Table 3 sensors-21-08010-t003:** Rates for real-life captures using full streams.

μ REDUCTION RATIO	84.73%
μ REDUCTION FACTOR	6.5
μ DETECTION RATE	99.83%

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
