# Peer review of "Application Layer Packet Processing Using PISA Switches"

_sensors, 2021, doi:10.3390/s21238010_

Round 1
Reviewer 1 Report
The paper is a good read, and all the units are clearly discussed in this article. The authors aimed to propose a method for PISA to process the application layer data. The motivation of this paper is very strong.
In fact, I have a few typographical errors as follows:
- When the P4 compiler is placed between the program and the API, what are the main functionalities of this compiler, specifically for the PISA switch?
- When the incoming flow arrives at the PISA switch, the control panel commands should be correctly integrated; authors can elaborate on this.
- The pseudo-code can be placed at the end of the article.
- The future research direction is very interesting, particularly the digital twin. Can the authors explain a little more on this point?
Reviewer 2 Report
This is an interesting study and the authors have investigated and proposed a solution for PISA switches in combination with application layer packet processing. The paper is generally well written and structured.
Organization of the paper is precise using exhibits proper grammar and punctuation. Technical quality and structure of the paper are clear, it meets the requirements well. Tables and figures are understandable and reinforce the narrative.
The use of statistics really helped in the interpretation of the results. Therefore the data analysis brings severity to the methodological approach.
Reviewer 3 Report
The topic Application Layer Packet Processing using PISA Switches is potentially interesting, however, there are some issues that should be addressed by the authors:
The Introduction" sections can be made much more impressive by highlighting your contributions. The contribution of the study should be explained simply and clearly.
The authors should further enlarge the Introduction with current work about machine and deep learning methods to improve the research background, for example: Packet-based network traffic classification using deep learning; Increasing web service availability by detecting application-layer DDoS attacks in encrypted traffic; Towards Secured Online Monitoring for Digitalized GIS Against Cyber-Attacks Based on IoT and Machine Learning.
The axes of figure 7 unclear
Clarify how you adjust the parameters of your technique
Conclusion section should be rearranged. According to the topic of the paper, the authors may propose some interesting problems as future work in the conclusion.
This study may be proposed for publication if it is addressed in the specified problems.
Round 2
Reviewer 3 Report
The authors handled all comments.